# Toughening Modification of Polylactic Acid by Thermoplastic Silicone Polyurethane Elastomer

**DOI:** 10.3390/polym13121953

**Published:** 2021-06-11

**Authors:** Mingtao Sun, Shuang Huang, Muhuo Yu, Keqing Han

**Affiliations:** College of Materials Science and Engineering, Donghua University, Shanghai 201620, China; mingtao-sun@foxmail.com (M.S.); dhuhuangshuang@163.com (S.H.); yumuhuo@dhu.edu.cn (M.Y.)

**Keywords:** polylactic acid, thermoplastic silicone polyurethane elastomer, toughening modification, blending modification, compatibility, mechanical properties

## Abstract

The melt blending of polylactic acid (PLA) and thermoplastic silicone polyurethane (TPSiU) elastomer was performed to toughen PLA. The molecular structure, crystallization, thermal properties, compatibility, mechanical properties and rheological properties of the PLA/TPSiU blends of different mass ratios (100/0, 95/5, 90/10, 85/15 and 80/20) were investigated. The results showed that TPSiU was effectively blended into PLA, but no chemical reaction occurred. The addition of TPSiU had no obvious effect on the glass transition temperature and melting temperature of PLA, but slightly reduced the crystallinity of PLA. The morphology and dynamic mechanical analysis results demonstrated the poor thermodynamic compatibility between PLA and TPSiU. Rheological behavior studies showed that PLA/TPSiU melt was typically pseudoplastic fluid. As the content of TPSiU increased, the apparent viscosity of PLA/TPSiU blends showed a trend of rising first and then falling. The addition of TPSiU had a significant effect on the mechanical properties of PLA/TPSiU blends. When the content of TPSiU was 15 wt%, the elongation at break of the PLA/TPSiU blend reached 22.3% (5.0 times that of pure PLA), and the impact strength reached 19.3 kJ/m^2^ (4.9 times that of pure PLA), suggesting the favorable toughening effect.

## 1. Introduction

Use of synthetic plastics derived from petroleum is challenged due to extremely well-known issues of white pollution. Seeking for renewable carbon resources as an alternative has become very important and urgent [1]. Polylactic acid (PLA) has been widely considered a potential alternative to replace conventional petroleum-based materials [2,3,4,5,6]. As a renewable resource derived from biomass with appropriate mechanical properties, good biocompatibility and degradability [7,8], PLA has experienced explosive market growth in biomedical materials [9,10,11], industrial packaging [12,13] and other short-time commodity applications [14,15,16]. However, its low heat resistance [17,18] and low toughness [19,20] severely limit its range of applications.

Several modification methods, such as copolymerization [20,21,22,23], plasticization [24,25,26,27] and blending with other polymers [28,29,30,31], have been proposed to toughen PLA. Copolymerization modification of PLA, such as poly(L-lactide-co-ε-caprolactone)(poly(CL-co-LA)) [23] and PLA-b-PEG [22], could toughen PLA efficiently and obtained favorable mechanical properties. However, the aforementioned PLA copolymers are uneconomical for wide application. And PLA was plasticized to increase flexibility, but the tensile modulus and tensile strength may decrease [32]. In addition, the plasticizers, such as low-molecular-weight poly(propylene glycol) [25] and poly(ethylene glycol) (PEG) [26], had a tendency to migrate from the bulk matrix to the surface, which caused the embrittlement of the blends. At present, the most practical and cost-effective method is blending modification [33,34]. Numerous polymers, such as linear low-density polyethylene [35,36], butylene adipate-terephthalate (PBAT) [37,38,39], poly(butylene succinate) [40,41,42,43] and natural rubber [44,45,46], etc., were blended with PLA to improve its toughness. But disappointingly, among these blends, many had poor comprehensive mechanical properties, reduced biodegradability or poor biocompatibility, which limited the applications of the blends. Some researchers have adopted the method of adding reactive compatibilizer [47,48] or a third component copolymer [49,50] to enhance the compatibility.

Thermoplastic polyurethane (TPU) is an elastomeric polymer with a unique combination of properties, such as high ductility, toughness, durability, flexibility, biocompatibility and biostability [51]. The molecular structure of TPU includes hard segments and soft segments. TPU obtains rigidity and hardness from the crystalline domain of the hard segment, while the amorphous soft segment provides its flexibility and elastic behavior [52,53,54]. Several studies explored how the tensile strength and elastic modulus of the blends made by TPU-toughened PLA gradually decreased with the increase of TPU content, while the elongation at break and impact strength were greatly increased [47,50,51,55,56,57,58,59]. Han et al. [58] prepared PLA/TPU blends by melt blending, and the mechanical test results showed that the elongation at break of the blend increased greatly, while the yield strength decreased. Feng and Ye studied the properties of PLA/TPU blends with different blending composition to explore an effective method for toughening of PLA [55]. The results showed that with the addition of TPU, brittle PLA changed to ductile material, and demonstrated that the blends were partially miscible because of the hydrogen bonding between the molecules of PLA and TPU. Similarly, Li [59] investigated the toughness of PLA/poly(ether)urethane (PLA/PU) blends. It was found that the blends were partially miscible system with shifted glass transition temperatures, revealed by dynamic mechanical analysis. Jing et al. [56] used TPU as a toughening modifier to melt blend with PLA, the results showed that the PLA/TPU blend was an incompatible system and had an obvious island structure. Nofar et al. [51] blended PLA with three different TPU grades with different hard segment content to study the effect of matrix crystallization on morphology and performance. Rheological experiments showed that the increase in hard segment content significantly improves the compatibility between PLA and TPU, although the use of TPU with lower hard segment was more conducive to enhancing the ductility and impact properties of the PLA.

Thermoplastic silicone polyurethane elastomer (TPSiU) is a copolymer which is obtained by silicone-modified thermoplastic polyurethane. Due to the introduction of silicone segments in the polyurethane structure, the Si–O bond and polysiloxane as the main body give TPSiU good flexibility and better heat resistance [60]. Therefore, compared with TPU, TPSiU has a better toughening effect on PLA on the one hand, and better heat resistance on the other hand. In this paper, we used TPSiU as the toughening agent, and prepared PLA/TPSiU blends by melt blending. The effects of TPSiU content on the molecular structure, crystallization, compatibility, thermal properties, mechanical properties and rheological properties of the blends were investigated.

## 2. Materials and Methods

### 2.1. Materials

Poly (lactic acid) (PLA) (Ingeo^TM^ 4032D) was supplied by (NatureWorks LLC, Minnetonka, Minnesota, US). The 4032D is a crystallizable grade of PLA with an L-lactide content of about 98.6 wt%. Lactic acid can be divided into L-lactic acid and D-lactic acid, which are enantiomers of each other. In the preparation process of L-lactic acid, the production of D-lactic acid was inevitable. Therefore, the PLA reported in this study had only 98.6 wt% L-lactide content, and the other 1.4 wt% component was D-lactide content, which did not affect the experimental results.

Thermoplastic silicone polyurethane elastomer (TPSiU, V170) was obtained from Meirui New Materials Ltd. (Shandong, China). 

### 2.2. Sample Preparation 

Before blending, PLA pellets were dried in a vacuum oven at 80 °C for 24 h and TPSiU pellets were dried at 80 °C for 6 h. The dried PLA and TPSiU were premixed in different PLA/TPSiU mass ratio (100/0, 95/5, 90/10, 85/15 and 80/20), and then melt blended by using a twin-screw extruder (TSE-186E, Nanjing Ruiya Polymer Equipment Company, Nanjing, China). The temperature of each section of the screw was 165 °C, 190 °C, 200 °C, 200 °C, 200 °C, and 180 °C in order, and the screw speed was 15–25 r/min. 

The blended pellets were dried in a vacuum oven at 80 °C for 24 h and then molded into samples for mechanical property testing with an injection-molding machine (SZ-5-C, Dehong Rubber and Plastic Machinery Company, Shanghai, China). The injection temperature was 165 °C, 230 °C, 230 °C, and 225 °C in order, and the mold temperature was 60 °C. The injection pressure was 30 MPa, injection speed was 2 cm/s, the pressure holding time was 8 s, and cooling time was 40 s.

### 2.3. Characterization

The crystallization and thermal properties were performed with a differential scanning calorimeter (Q20, TA Company, New Castle, Delaware, US). 

The amount of TPSiU sample was 5–8 mg. In a nitrogen atmosphere, the flow rate was 50 mL/min for testing. The sample was heated from room temperature to 250 °C at a heating rate of 20 °C/min to eliminate thermal history. After a constant temperature for 3 min, it was reduced to –75 °C at a rate of 10 °C/min. Finally, it was heated to 250 °C at a heating rate of 10 °C/min and recorded the cooling curve and the second heating curve.

PLA/TPSiU samples of about 5–8 mg were heated from room temperature to 200 °C at a heating rate of 20 °C/min under nitrogen atmosphere and hold for 3 min to eliminate thermal history, and then annealed to room temperature at a rate of 10 °C/min, followed by heating to 200 °C at a heating rate of 10 °C/min. The second heating curves were recorded.

The crystallinity of the blends was calculated using Equation (1) [9]:(1)$$X_{c}=\frac{\Delta \mathrm{Hm}-\Delta \mathrm{Hc}}{f\;\times \;93.6}$$
where ΔHm is the enthalpy of melting, ΔHc is the enthalpy of cold crystallization, f is mass fraction of PLA and 93.6 is melting enthalpy for 100% crystalline PLA.

Thermogravimetric analysis (TGA) was carried out using a thermogravimetric analyzer (TG 209 F1, NETZSCH Company, Selb, Germany). About 5 mg of each sample was heated from room temperature to 500 °C in a heating rate of 10 °C/min with a nitrogen flow rate of 20 mL/min. 

The chemical functional groups of PLA, TPSiU and PLA/TPSiU blends were investigated by a Fourier transform infrared (FTIR) spectrometer (Nicolet 6700, Nicolet Company, Madison, Wisconsin, US) in the range of 600–4000 cm^−1^ at a resolution of 4 cm^−1^. The samples were thoroughly dried before FTIR measurements. After drying, using the OMNI sampler single attenuated total reflection method, the spectra for samples of PLA/TPSiU were recorded.

Dynamic mechanical analysis (DMA) was performed with a TA Q800 (New Castle, Delaware, US) instrument in a single cantilever beam mode. The dynamic loss (tanδ) and the storage modulus (E′) were determined at a frequency of 1 Hz and a heating rate of 3 °C/min as a function of temperature from –100 °C to 150 °C. The 36 × 13 × 3 mm^3^ samples were prepared by injection molding.

Tensile tests were carried out according to ISO 527-4:1997 with a universal material testing machine (WDW3020, Kexin Test Instrument Company, Changchun, China) at a loading speed of 5 mm/min. At least 6 specimens were tested for each sample to obtain an average value. Figure 1 shows the size of the stretched spline.

The pendulum impact tester (JUD-50, Juyuan Testing Equipment Company, Chengde, China) was applied to test the notched impact strength of PLA/TPSiU with a V-notch depth of 2.54 mm according to ISO 180:2000. At least five specimens were tested for each sample. 

The morphology of the blends was observed by a scanning electron microscope (Quanta 250, FEI Company, Brno, Czech Republic) at 10 kV, using a magnification of 5000×. The samples were fractured in liquid nitrogen, and then coated with a thin layer of gold before examination.

Capillary rheometer (RH2000, Malvern Company, Malvern, UK) was used to determine the rheology of the PLA/TPSiU blend system. The diameter of the capillary was 0.5 mm, the ratio of length to diameter was 8:1, and the shear rate was in the range of 100–5500 s^−1^. The sample should be vacuum-dried for 12 h at 80 °C before testing.

## 3. Results and Discussion

### 3.1. Molecular Structure

Figure 2 showed the infrared spectra of pure PLA, PLA/10 wt%TPSiU, PLA/15 wt%TPSiU and TPSiU. The analysis of the TPSiU infrared spectrum was as follows: the N–H stretching vibration peak of thermoplastic polyurethane elastomer (TPU) was at 3500–3100 cm^−1^, and the C=O stretching vibration peak was at 1800–1650 cm^−1^. Moreover, the N–H in the hard segment of the macromolecule can form intermolecular hydrogen bonds with C=O in the hard segment and –O– or C=O in the soft segment. TPSiU had a strong absorption peak at 3303 cm^−1^, which was the N–H stretching vibration peak after forming hydrogen bonds with hard carbamate. The C=O stretching vibration peaks at 1731 cm^−1^ and 1704 cm^−1^, the absorption peak at 1731 cm^−1^ was the carbonyl group that has not formed a hydrogen bond, and the absorption peak at 1704 cm^−1^ was the carbonyl group that has formed a hydrogen bond. C–H stretching vibration peaks were at 2919 cm^−1^ and 2851 cm^−1^, C=O stretching vibration peaks in urea were at 1640 cm^−1^, NH(–C–NH-) bending vibration peaks of amide near 1533 cm^−1^. Near 1224 cm^−1^ was the C–O stretching vibration peak in the ester group [61]. The strong absorption peak at 1108 cm^−1^ and the shoulder peak at 1020 cm^−1^ were attributed to the asymmetric stretching vibration of C–O–C in the soft segment of TPSiU and the overlap of the characteristic peaks of Si–O–Si. The absorption peak at 810 cm^−1^ was Si-CH_3_ stretching vibration. According to the above analysis, the main chain of TPSiU contains not only the typical structure of TPU, but also the Si–O–Si structure in silicone.

For pure PLA, the CH_3_ asymmetric stretching and asymmetric deformation modes appeared at about 2997 and 1455 cm^−1^, respectively. The band at about 1755 cm^−1^ was the vibration of C=O. The bands at 1182, 1131, and 1086 cm^−1^ were related to the C–O–C stretching vibration modes. With the addition of TPSiU, PLA/TPSiU blends had almost the same FTIR spectra with pure PLA. It is worth noting that the bending vibration peak of N–H in -NHCOO- appeared near 1533 cm^−1^ and the stretching vibration peak of Si-CH_3_ appeared at 810 cm^−1^, which provided the evidence that TPSiU was effectively blended into PLA, but no chemical reactions occurred.

### 3.2. Crystallization

Figure 3a,b showed the cooling curve and secondary heating curve of TPSiU, respectively. It can be seen from that the glass transition temperature of TPSiU was not obvious, and there was no obvious crystallization exothermic peak and melting endothermic peak in the whole range, indicating that TPSiU was an amorphous polymer.

Figure 4 showed differential scanning calorimetry (DSC) secondary heating curves of PLA/TPSiU blends. Table 1 lists the thermal parameters of PLA/TPSiU blends derived from Figure 4. It can be seen from Table 1 that the difference between the cold crystallization enthalpy (ΔHc) and the melting enthalpy (ΔHm) of PLA/TPSiU blends were slight, indicating that the crystallization ability of PLA/TPSiU blends was limited during the cooling process, and the crystallinity of PLA/TPSiU blends was less than 10 wt%, which was still close to an amorphous structure. With the addition of TPSiU, the cold crystallization peak of the blend moved to high temperature and the crystallinity decreased slightly, suggesting that the addition of TPSiU hindered the crystallization of PLA. This may be due to the formation of hydrogen bond between the C=O groups in the PLA molecular structure and the N–H groups in TPSiU, which interfered the motion of PLA molecular chain during the crystallization process, and therefore the crystallization performance decreased. 

### 3.3. Thermal Properties

The introduction of siloxane structure into polyurethane can effectively improve its heat resistance. This is mainly due to the bond energy of the Si–O bond (452 kJ/mol) in the siloxane structure is much greater than the bond energy of the C–C bond (345 kJ/mol) and C–O bond (351 kJ/mol). Therefore, the heat resistance of the polyurethane incorporated with the silicone segment is improved.

Figure 5a,b showed the TGA curve and DTG curve of TPSiU. The thermodynamics of TPSiU were mainly divided into two weight loss sections: 275–380 °C and 380–470 °C. The weight loss rate of the first stage was about 29%. This was due to the decomposition of the hard urethane in the TPSiU molecular structure. The thermal degradation temperature of the urethane was lower than that of the urea group, so the carbamate was degraded first. The weight loss rate of the second stage was about 59%, which corresponds to the decomposition of the polyester and silicone skeleton in the TPSiU soft stage. In the whole thermal degradation process, the hard segment had the fastest degradation rate at 340 °C, and the soft segment had the fastest degradation rate at 415 °C. The temperature of 5 wt% weight loss corresponds to 320 °C, indicating that the product had good thermal stability.

TGA curves and derivative thermogravimetric (DTG) curves of PLA/TPSiU blends under nitrogen atmosphere were shown in Figure 6a,b, respectively. The initial decomposition temperature(T_5%_), defined as the temperature at 5 wt% weight loss, and the final decomposition temperature (T_end_) of PLA/TPSiU blends were summarized in Table 2. T_5%_ of PLA/TPSiU blends decreased slightly with the increase of TPSiU content. When the TPSiU content reached above 15 wt%, T_5%_ of PLA/TPSiU blends decreased about 10 °C compared with pure PLA. This may be due to the T_5%_ of TPSiU being lower than that of pure PLA. The fastest weight loss rate temperature (T_max_) of all samples were almost similar. The enlarged curves in the 400–480 °C region were shown in the inset of Figure 6a. In the inset, one can clearly identify that T_end_ of PLA/TPSiU blends was improved significantly. It is also exhibited that the residual mass at 500 °C increased when TPSiU content increased, which resulted from the non-decomposable silicon contained in TPSiU. From the above results, the thermal stability of PLA/TPSiU blends decreased slightly with the increase of TPSiU content, but the T_5%_ of the blend system was in the range of 320–330 °C, and the T_max_ was in the range of 365–367 °C.

### 3.4. Compatiblity

Dynamic mechanical properties in the solid state were investigated employing the flat sheets obtained by the injection molding. The dependence of storage modulus E′ for PLA/TPSiU blends with different TPSiU content was shown in Figure 7. The storage modulus E′ of PLA decreased sharply at around 62–80 °C, indicating that the molecular segments obtained enough thermal energy to continuously change the conformation and realized the transition from the glass state to the rubber state. As for PLA/TPSiU blends, the peak temperature due to the glass transition of PLA was independent of the blend ratio, suggesting that mutual dissolution did not occur. The storage modulus E′ below the glass transition temperature of the blends decreased compared with the pure PLA. Also, the increase of TPSiU content caused a distinct decrease in the storage modulus E′ because the storage modulus of TPSiU was much lower than that of pure PLA as shown in Figure 7.

Figure 8 presents the plots of tanδ against temperature for PLA/TPSiU blends. It can be seen from Figure 8 that both pure PLA and TPSiU had one peak, corresponding to their glass transition temperature respectively. From the enlarged illustration, two peaks appeared between −100 and 100 °C for the PLA/TPSiU blends. Usually, the glass transition peak of polymer blends approached each other if the compatibility was improved in some level. In this study, the glass transition peak (near −55 °C) for TPSiU was slightly enhanced with the increase of TPSiU content, suggesting that there was a clear phase separation in the blends at the micro level. When the TPSiU content increased from 10 wt% to 20 wt%, the glass transition temperature corresponding to PLA and TPSiU in the blends changed slightly, indicating that the two components had poor thermodynamic compatibility. This was like the results of Jing et al. [56] using DMA to characterize the compatibility of PLA/TPSiU. Nofar et al. [51] used TPU with high hard segment content to toughen PLA. The results showed that TPU with high hard segment content has better phase compatibility between PLA and TPU, although the use of lower hard segments is more favorable to enhance the ductility and impact properties of the blends.

### 3.5. Mechanical Properties

The stress–strain curves of PLA/TPSiU blends were shown in Figure 9. Table 3 is elongation at break for PLA/TPSiU blends. It can be seen from the Figure 9 that there was no yield phenomenon when pure PLA was stretched, and the elongation at break was only 4.5%, which showed brittle fracture. The elongation at break of TPSiU was 900% (data provided by TPSiU manufacturer). When TPSiU was added, the tensile behavior of PLA had undergone significant changes, and the blends had obvious yielding and plastic deformation platforms during the stretching process, which was a typical ductile fracture. The elongation at break of the blends gradually increased with the increase of TPSiU content, and the “neck” appeared during the stretching process. This was consistent with the change trend of elongation at break in the study of PLA/TPU blends by Fei Feng et al. [55]. When the addition amount of TPSiU was 15 wt%, the elongation at break of the blend was 22.3%, which is pure PLA about 5.0 times.

Figure 10 shows the tensile strength and tensile modulus of PLA/TPSiU blends. The tensile strength of TPSiU was 13 MPa (data provided by TPSiU manufacturer). It was found that the tensile strength of PLA/TPSiU blends decreased gradually with the increase of TPSiU content, which was consistent with previous research results on rubber-toughened polylactic acid [44,45]. On the one hand, this may be due to the existence of TPSiU, which weakened the interaction between the molecular chains of PLA, leading to premature breakage during the stretching process. On the other hand, due to the elastomeric nature of TPSiU, its inherent tensile strength was lower than that of PLA, resulting in a decrease in the tensile strength of the blends. When the amount of TPSiU added was 15 wt%, the tensile strength was 74.5% of pure PLA, and the decrease rate increased with the increase of the TPSiU content. When the content of TPSiU was 20 wt%, the tensile strength was about 43.1 MPa, only 65.4% of pure PLA. In comparison with other studies, the remarkably improved impact toughness is commonly accompanied by weakened tensile strength, that is, up to 50% of decrease compared to pristine PLA [30]. Therefore, when blending and toughening modified polylactic acid is used, it is necessary to fully consider the comprehensive mechanical properties while improving its toughness.

The tensile modulus of PLA/TPSiU blends was also measured for various TPSiU content as shown in Figure 10. When the content of TPSiU was less than 15 wt%, the tensile modulus of PLA/TPSiU blends was almost equivalent to that of pure PLA, of about 1.7 GPa. Once the TPSiU content exceeded 20 wt%, the tensile modulus of the blends dropped to 1.2 GPa rapidly. This may be attributed to the fact that the tensile modulus of TPSiU was much lower than that of PLA. Also, the uniform dispersion of TPSiU in the matrix was difficult when the content of TPSiU increased to 20 wt%, which led to poor compatibility, and the debonding occurred at the TPSiU–PLA interface owing to the insufficient interfacial adhesion and, therefore the tensile modulus decreased.

The notched impact strength of PLA/TPSiU blends was shown in Figure 11. As a typical brittle material, the notched impact strength of pure PLA was only 3.9 kJ/m^2^. The addition of TPSiU significantly improved the notched impact strength of PLA/TPSiU blends. When the content of TPSiU was 15 wt%, the impact strength of the blend increased to 19.3 kJ/m^2^, which was 4.9 times that of pure PLA, implying the favorable toughening effect of TPSiU on PLA. It is considered that the TPSiU domains acted as stress concentrators upon being subjected to the impact test. Owing to the elastic property, TPSiU domains could deform to absorb the impact energy, therefore improving the impact toughness of the blends. Meanwhile, as the content of TPSiU increased, the number of dispersed TPSiU domains, and the rate of initiation, branching and termination of crazing, increased which resulted in improvement in the impact strength of the blends accordingly. When the content of TPSiU reached 20 wt%, the impact strength of the PLA/TPSiU blends decreased slightly, which may have been caused by the uneven dispersion of TPSiU in the PLA matrix and the larger size of domains.

### 3.6. Morphology

Figure 12 shows scanning electron microscopy (SEM) images of cross section of pure PLA and PLA/TPSiU blends. Pure PLA presents a smooth and flat morphology, showing typical brittle fracture. PLA/TPSiU blends had a rough cross section and obvious plastic deformation, especially with the increase of TPSiU content. Moreover, the cross section of PLA/TPSiU blends displayed a clear sea-island structure, in which the island phase was TPSiU even if uniformly dispersed in the matrix, suggesting that the compatibility between the two phases were poor. When the TPSiU content was 5 wt%, the size of the “island” phase was 1.3~3.3 µm. With the increase of TPSiU content, the size of the “island” phase gradually increased, and the two-phase structure was more obvious.

When 5 wt% TPSiU was added, stress whitening occurs on the quenched section of the blend, and it was most obvious when the addition content was 15 wt%. This may be due to the fact that TPSiU acts as a stress concentrator to induce a lot of crazing and shear bands. The generation and development of the shear zone consumes a lot of energy and delays the destruction process of the material, thus significantly improving the toughness of the blends. When the content of TPSiU increased to 20 wt%, the quenched section of the blend was smoother than that of the blend with a TPSiU content of 15 wt%, and many voids were distributed in the matrix. This may be due to the excessive content and larger size of TPSiU. The bonding force between the two interfaces becomes smaller, and the dispersed phase falls off from the matrix when subjected to external force.

### 3.7. Rheology

Figure 13a,b showed the relationship between apparent viscosity and shear rate of PLA/TPSiU blends with different ratios at 230 °C and 240 °C, respectively. First of all, it can be seen from Figure 13 that the apparent viscosity of PLA/TPSiU blends with different proportions gradually decreases with the increase of shear rate, which is a typical characteristic of pseudoplastic fluids.

Secondly, when a small amount of TPSiU (5 wt%) was added, the apparent viscosity of the melt was slightly higher than that of pure PLA. As the amount of TPSiU increases, the melt viscosity of the blend system gradually decreases. The increase in viscosity was because when a small amount of TPSiU (5 wt%) was added, the interaction between PLA and TPSiU molecular chains was enhanced, resulting in increased entanglement between molecular chains and increased resistance to molecular chain movement. As the content of TPSiU increased, the melt viscosity decreased, which may have been due to the lower molecular weight TPSiU acting as a plasticizer, increasing the free volume of the blending system, weakening the intermolecular forces, making the melt easier to flow, and reducing the melt viscosity.

## 4. Conclusions

In this paper, PLA was physical blended and toughened by TPSiU. The results showed that when 15% TPSiU was added, the elongation at break of the PLA/TPSiU blend reached 22.3% (5.0 times that of pure PLA), and the impact strength reached 19.3 kJ/m^2^ (4.9 times that of pure PLA). It is worth mentioning that while TPSiU exhibited a favorable toughening effect, the tensile strength of the PLA/TPSiU was maintained at 75% of that of pure PLA, the tensile modulus remained basically unchanged, and the comprehensive mechanical properties were maintained. The thermal stability of PLA/TPSiU blends changed little. However, the blends exhibited typical island-in-sea phase separation suggesting the poor compatibility, which was also demonstrated by DMA results.

## Figures and Tables

**Figure 1 polymers-13-01953-f001:**
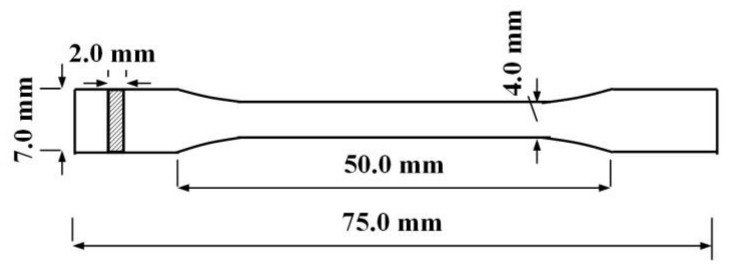
Stretching spline size chart.

**Figure 2 polymers-13-01953-f002:**
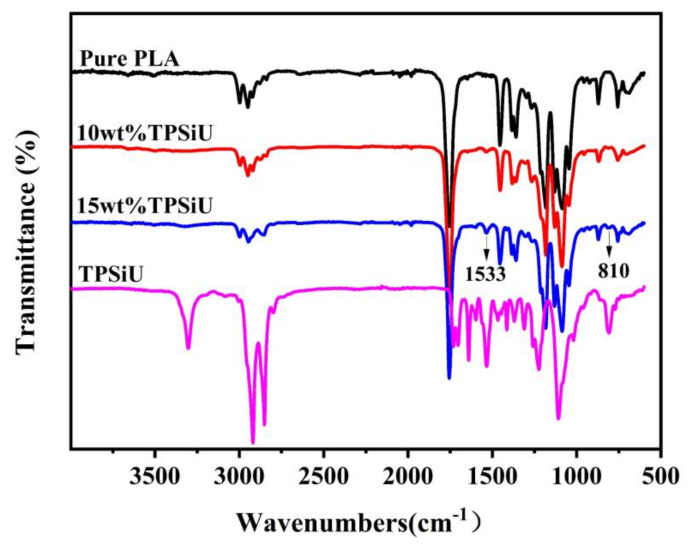
Fourier transform infrared (FTIR) spectra for polylactic acid (PLA), PLA/10 wt% thermoplastic silicone polyurethane elastomer (TPSiU), PLA/15 wt% TPSiU and TPSiU.

**Figure 3 polymers-13-01953-f003:**
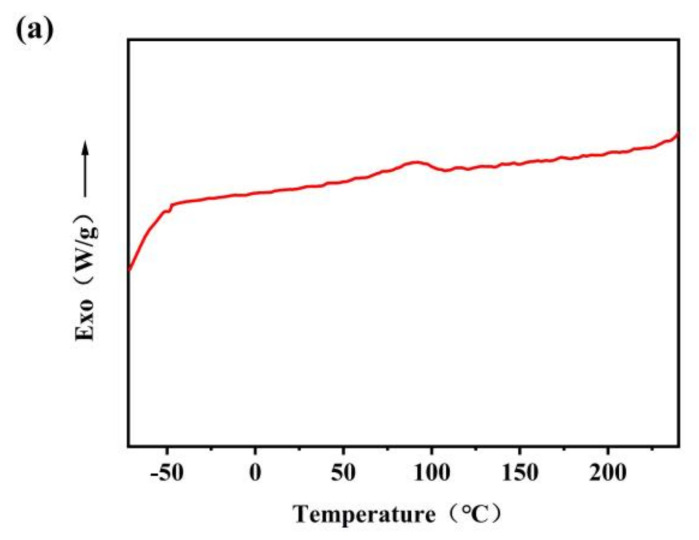
Differential scanning calorimetry (DSC) curves of TPSiU. (**a**) cooling curve. (**b**) secondary heating curve.

**Figure 4 polymers-13-01953-f004:**
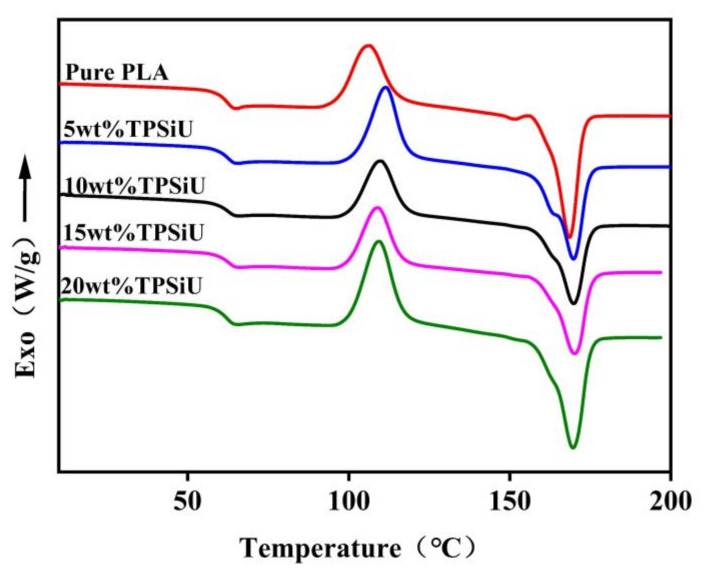
DSC heating curves of PLA/TPSiU blends.

**Figure 5 polymers-13-01953-f005:**
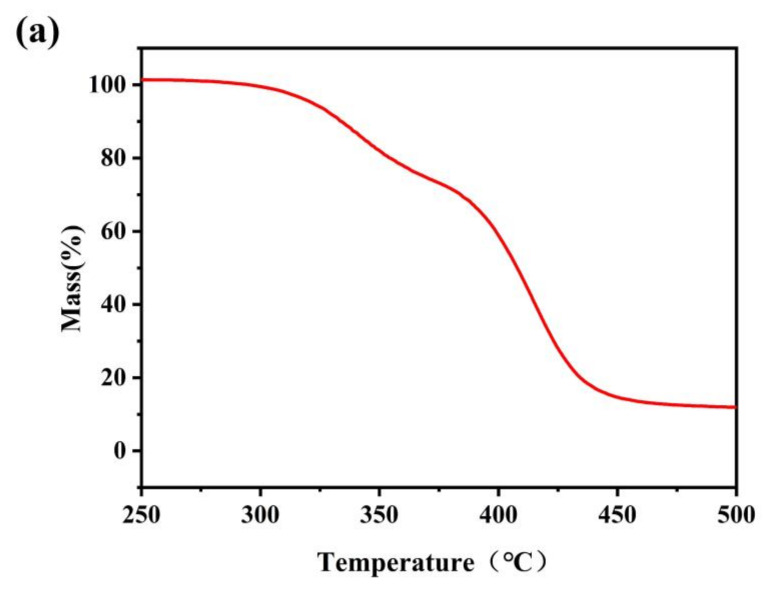
Thermogravimetric analysis (TGA) curve of TPSiU (**a**) and corresponding derivative thermogravimetric (DTG) curve (**b**).

**Figure 6 polymers-13-01953-f006:**
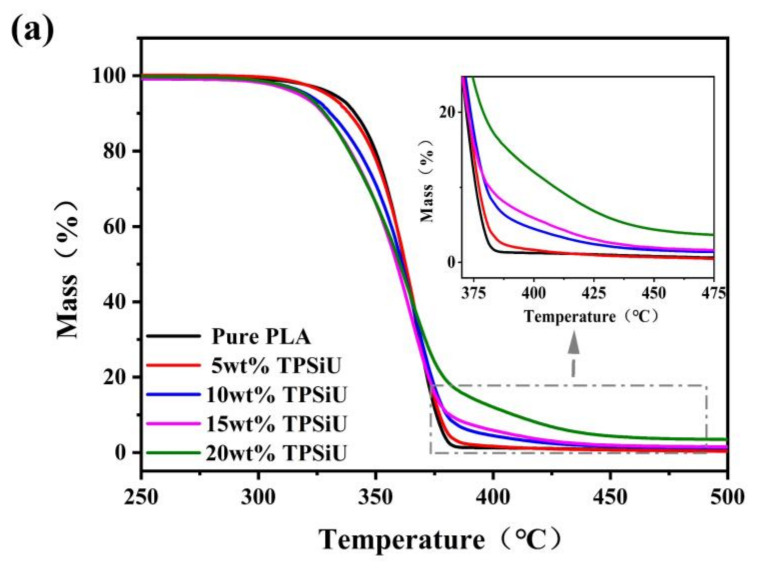
TGA curves of PLA/TPSiU blends (**a**) and corresponding DTG curves (**b**).

**Figure 7 polymers-13-01953-f007:**
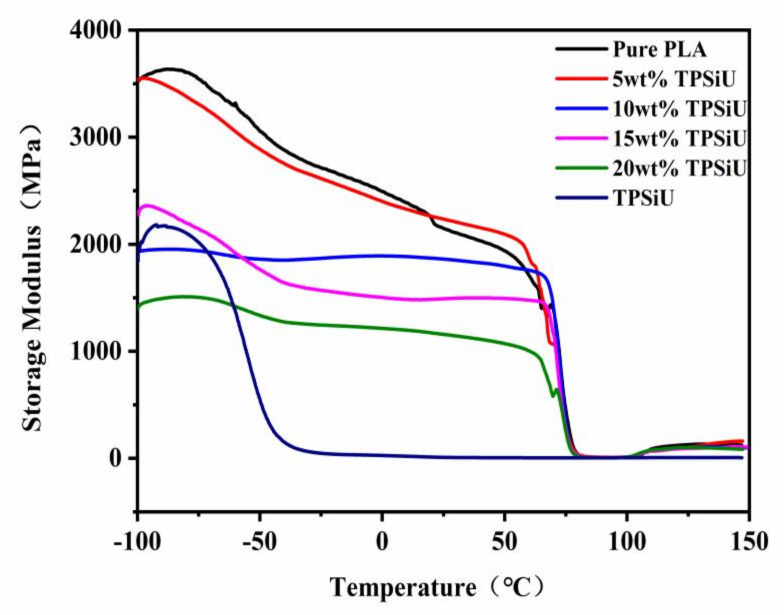
Temperature dependence of storage modulus E′ for PLA/TPSiU blends.

**Figure 8 polymers-13-01953-f008:**
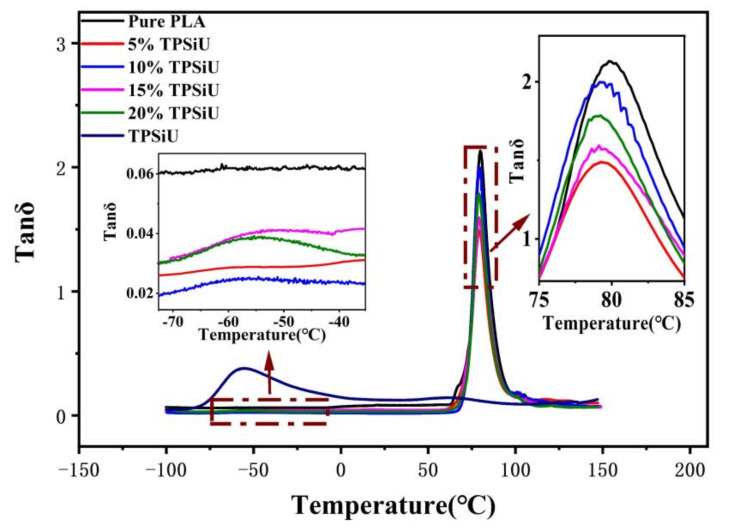
Temperature dependence of tan δ for PLA/TPSiU blends.

**Figure 9 polymers-13-01953-f009:**
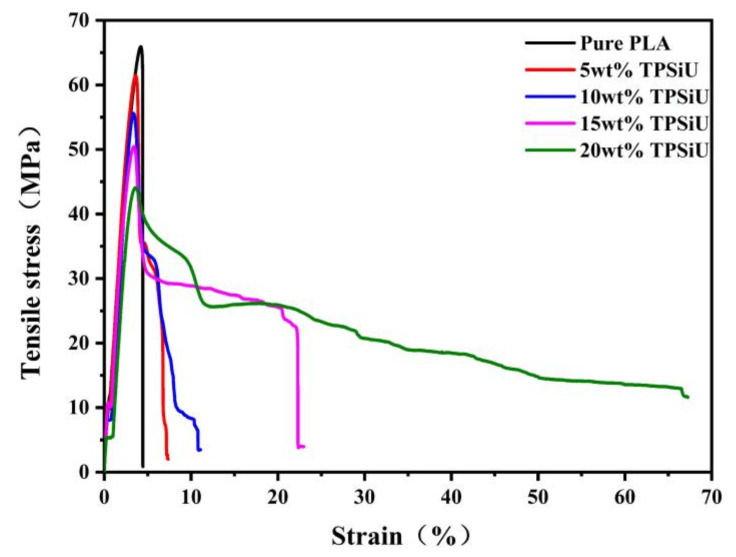
Stress–strain curves of PLA/TPSiU blends.

**Figure 10 polymers-13-01953-f010:**
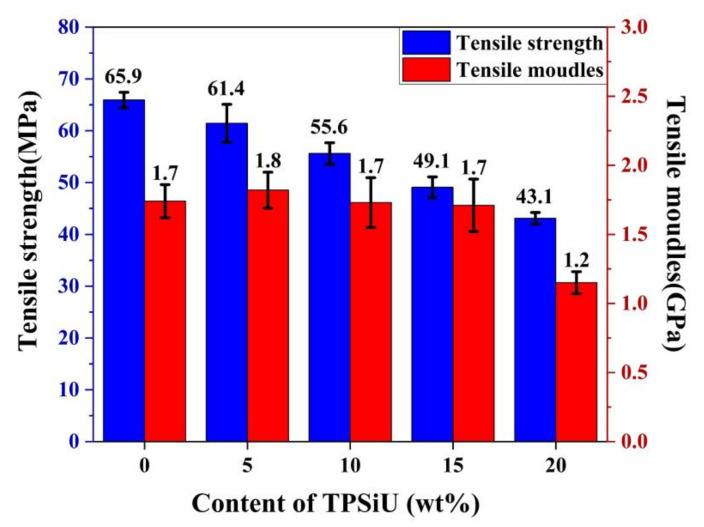
Tensile strength and tensile modulus for PLA/TPSiU blends.

**Figure 11 polymers-13-01953-f011:**
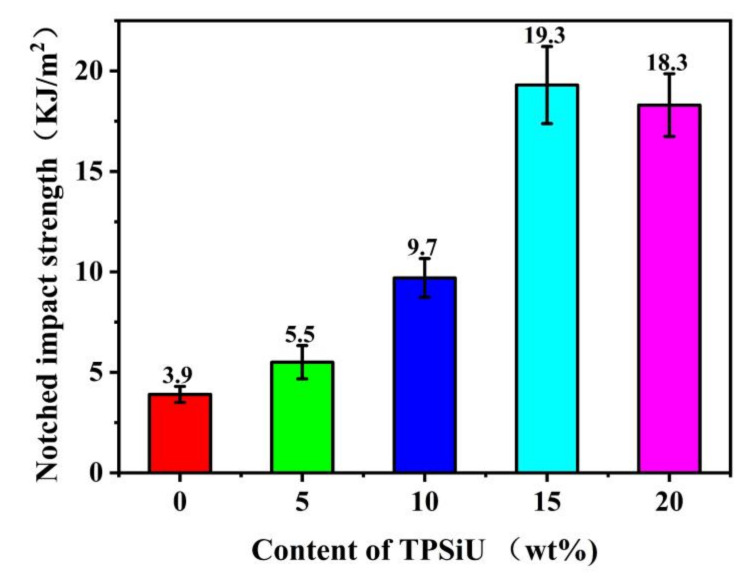
Effects of TPSiU content on the notched impact strength of PLA/TPSiU blends.

**Figure 12 polymers-13-01953-f012:**
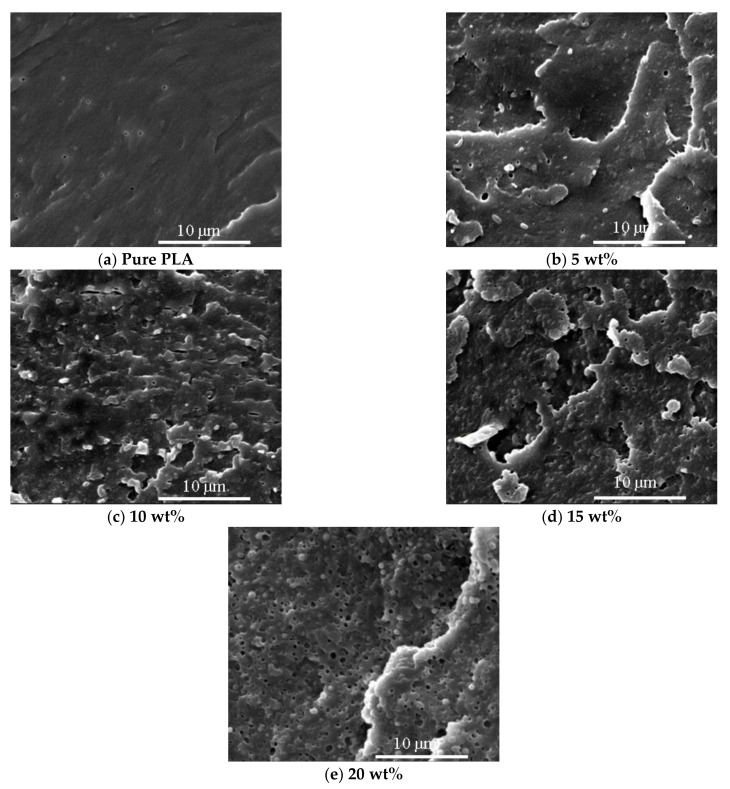
Scanning electron microscopy (SEM) photographs of cross section of PLA/TPSiU blends with different TPSiU content.

**Figure 13 polymers-13-01953-f013:**
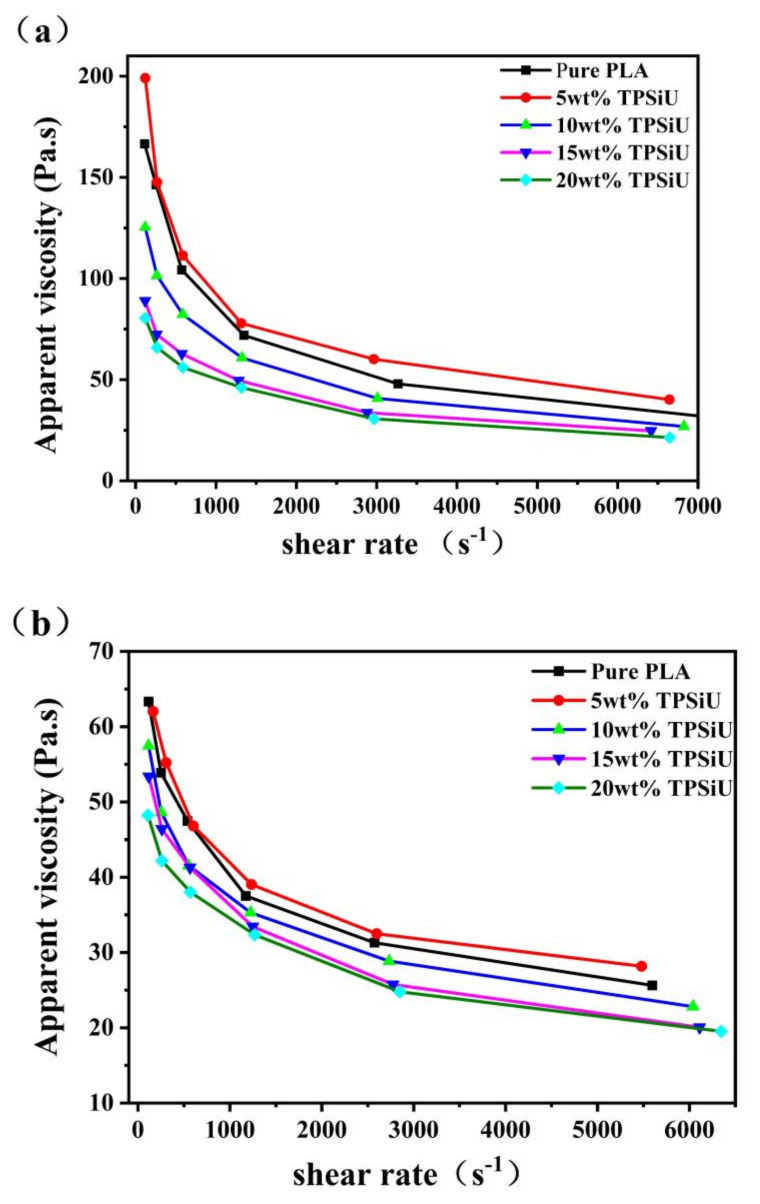
Apparent viscosity-shear rate curves of PLA/TPSiU blends, (**a**) 230 °C, (**b**) 240 °C.

**Table 1 polymers-13-01953-t001:** Thermal parameters of PLA/TPSiU blends according to DSC.

PLA/TPSiU (*w*/*w*)	Tg (°C)	Tc (°C)	ΔHc (Jg^−1^)	ΔHm (Jg^−1^)	Tm (°C)	Crystallinity Xc(%)
100/0	62.5	106.2	25.2	31.6	162.1	6.8
95/5	62.8	111.4	26.1	30.4	169.7	4.8
90/10	62.3	109.8	21.7	26.5	169.8	5.7
85/15	62.8	109.0	20.7	24.8	170.1	5.1
80/20	62.2	109.3	27.6	32.1	169.7	6.1

**Table 2 polymers-13-01953-t002:** TGA parameters for PLA/TPSiU blends.

PLA/TPSiU (*w*/*w*)	100/0	95/5	90/10	85/15	80/20
T_5%_ (°C)	332	329	321	317	320
T_max_ (°C)	365	366	367	365	366
T_end_ (°C)	386	391	438	472	478

**Table 3 polymers-13-01953-t003:** Elongation at break for PLA/TPSiU blends.

PLA/TPSiU (*w*/*w*)	100/0	95/5	90/10	85/15	80/20
Elongation at break (%)	4.5 ± 0.33	7.3 ± 0.17	11.9 ± 0.75	22.3 ± 1.62	66.0 ± 2.53

## Data Availability

Data are contained within the article.

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
