# Peer review of "Toughening Modification of Polylactic Acid by Thermoplastic Silicone Polyurethane Elastomer"

_polymers, 2021, doi:10.3390/polym13121953_

Round 1
Reviewer 1 Report
Reviewers' comments:
Manuscript Number: polymers-1243348
Full Title: Toughening Modification of Polylactic acid by Thermoplastic Silicone Polyurethane Elastomer.
Comments:
The manuscript reported on Toughening Modification of Polylactic acid by Thermoplastic Silicone Polyurethane Elastomer. The manuscript needs a detailed editing. It cannot be recommended for publication in the present form. I hope the following points would be helpful for the authors.
- Qualitative information’s are missing in abstract.
- Add more keywords.
- The introduction is very poor and less informative. Authors should elaborate their introduction section by citing few more relevant references. The novelty of the work should also be highlighted.
- The experimental section should be detailed.
- In part SEM: how the energy of the accelerator beam used?
- 3.3 Molecular structure and 3.6 Morphology - should be detailed
- Figure 8, is not clear make clear.
- Conclusion should be concise.
- References: there are recent references in 2020 and 2021 treating the same subject, you can use and make all references in same format for volume number, page numbers and journal name, because it is difficult to searching and reading.
- Language needs substantial improvement. Please consult a native English speaker or a language editing service.
Based on these, I advise the authors to rectify the above mentioned errors and we hope to re-evaluate the revised manuscript.
Author Response
Response to Reviewer 1 Comments
Point 1: (Qualitative information’s are missing in abstract.)
Response 1: As the reviewer suggested, we revised the abstract.
Point 2: (Add more keywords)
Response 2: We have made supplements based on the comments of the reviewer.
Point 3: (The introduction is very poor and less informative. Authors should elaborate their introduction section by citing few more relevant references. The novelty of the work should also be highlighted.)
Response 3: We have re-written this part according to the reviewer’s suggestion. We are very sorry for our negligence of the reasons for choosing TPSiU. Polyurethane had been proven to have a better toughening effect. Due to the introduction of silicone segments in the polyurethane structure, the Si-O bond and polysiloxane as the main body which makes TPSiU have good flexibility and more excellent heat resistance. Therefore, compared with TPU, TPSiU has better toughening effect on PLA on the one hand, and better heat resistance on the other hand. And the experimental results obtained in the paper also proved this point. So we choosed TPSiU to plasticize PLA.
Point 4: (The experimental section should be detailed)
Response 4: We have made correction according to the reviewer’s comments.
Point 5: (In part SEM: how the energy of the accelerator beam used?)
Response 5: We have made correction according to the reviewer’s comments. Accelerating voltage was 10 kV, using a magnification of 5000×.
Point 6: (3.3 Molecular structure and 3.6 Morphology - should be detailed)
Response 6: It is really true as reviewer suggested that molecular structure and morphology should be detailed. We have made corrections and supplements based on the reviewers’ comments.
Point 7: (Figure 8, is not clear make clear)
Response 7: As suggested by the reviewer, we adjusted the electron microscope picture, And hope that the correction will meet with approval.
Point 8: (Conclusion should be concise)
Response 8: We have re-written this part according to the reviewer’s suggestion.
Point 9: (References: there are recent references in 2020 and 2021 treating the same subject, you can use and make all references in same format for volume number, page numbers and journal name, because it is difficult to searching and reading.)
Response 9: We have made correction according to the reviewer’s comments. We have cited some recent literature in the introduction to make the paper more substantial. And checked the format of the paper citations to meet the journal's requirements.
Point 10: (Language needs substantial improvement. Please consult a native English speaker or a language editing service.)
Response 10: Taking into account the reviewer's suggestions, we carefully reviewed and revised the paper. And hope that the correction will meet with approval.
Other changes:
We have made corrections and supplements based on the reviewers’ comments in order to make the paper more complete. we have added some experimental data and revised the planning sequence of the Section 3 of the paper.
We have studied reviewer’s comments carefully . We have tried our best to revise our manuscript according to the comments. Attached please find the revised version, which we would like to submit for your kind consideration.
We would like to express our great appreciation to you and reviewers for comments on our paper. Looking forward to hearing from you.
Thank you and best regards.

Reviewer 2 Report
In line 133. There is a missing space on the sentence about Table 1.
In line 184. There exist a grammatical error in the word dereased, must be: decreased.
In general, the results are poorly supported by the discussion, besides exists a lack of comparison of their results respect of the literature reports. Section 3, must be enhanced.
Author Response
Response to Reviewer 2 Comments
Point 1: There is a missing space on the sentence about Table 1
Response 1: We are very sorry for our incorrect writing. And we have made correction according to the Reviewer’s comments.
Point 2: There exist a grammatical error in the word dereased, must be: decreased.
Response 2: As reviewer suggested that a grammatical error in the paper, And we have made correction according to the Reviewer’s comments.
Point 3: In general, the results are poorly supported by the discussion, besides exists a lack of comparison of their results respect of the literature reports. Section 3, must be enhanced.
Response 3: As reviewer suggested, We have re-written this part according to the reviewer’s suggestion and added some experimental data. Due to the large size of the revised paper, it is not listed here. Please see the attachment.
Other changes:
We have made corrections and supplements based on the reviewers’ comments in order to make the paper more complete. we have added some experimental data and revised the planning sequence of the third part of the paper.
We have studied reviewer’s comments carefully . We have tried our best to revise our manuscript according to the comments. Attached please find the revised version, which we would like to submit for your kind consideration.
We would like to express our great appreciation to you and reviewers for comments on our paper. Looking forward to hearing from you.

Reviewer 3 Report
Dear authors
I have overall enjoyed article reading. The manuscript is properly structured and the quality of presentation is remarkable. However, the results are not promising. As mentioned in the conclusion section, the addition of the TPSiU improved only some the mechanical properties when compared to pure PLA. And also, some properties deteriorated. Based on the experimental results, could you please specify why TPSiU was chosen for this study? Some other concerns are next listed:
Line 79: It called my attention that the PLA reported in this research has only 98.6 wt% content of L-lactide. What is the other 2.4 wt% content made of? Bear in mind that the TPSiU was mixed with PLA in a 95/5 mass ratio, so the existence of “other compounds” in the PLA is comparable with the TPSiU. Do you expect that this 2.4 wt% content of “other compounds” to play a role in the experimental results of your paper?
Figure 4: When looking at this chart, I wondered why TPSiU was chosen for blending with the PLA? As you mentioned, the TPSiU has a lower storage modulus than PLA, and therefore, every PLA/ TPSiU blend resulted in lower storage modulus than the pure PLA. What is the net advantage of mixing PLA with TPSiU?
Minor corrections are next listed:
Line 87: Could you give some further details regarding the time employed for the melt blend and injection-molding processes? Was extrusion performed only once?
Figure 1: What are the y-axis units of this figure?
Author Response
Response to Reviewer 3 Comments
Point 1: Based on the experimental results, could you please specify why TPSiU was chosen for this study?
Response 1: We are very sorry for our negligence of the reasons for choosing TPSiU. Polyurethane had been proven to have a better toughening effect. Due to the introduction of silicone segments in the polyurethane structure, the Si-O bond and polysiloxane as the main body which makes TPSiU have good flexibility and more excellent heat resistance. Therefore, compared with TPU, TPSiU has better toughening effect on PLA on the one hand, and better heat resistance on the other hand. And the experimental results obtained in the paper also proved this point. So we choosed TPSiU to plasticize PLA.
Point 2: It called my attention that the PLA reported in this research has only 98.6 wt% content of L-lactide. What is the other 2.4 wt% content made of? Bear in mind that the TPSiU was mixed with PLA in a 95/5 mass ratio, so the existence of “other compounds” in the PLA is comparable with the TPSiU. Do you expect that this 2.4 wt% content of “other compounds” to play a role in the experimental results of your paper?
Response 2: Lactic acid can be divided into L-lactic acid and D-lactic acid, which are enantiomers of each other. The current mainstream method of synthesizing polylactic acid is to synthesize lactide from lactic acid first, and then form polylactic acid by ring-opening polymerization of lactide. In the preparation process of L-lactic acid, the production of D-lactic acid is inevitable. Therefore, the PLA reported in this study has only 98.6 wt% L-lactide content, and the other 1.4 wt% component is D-lactide content, which will not affect the experimental results.
Point 3: When looking at this chart, I wondered why TPSiU was chosen for blending with the PLA? As you mentioned, the TPSiU has a lower storage modulus than PLA, and therefore, every PLA/ TPSiU blend resulted in lower storage modulus than the pure PLA. What is the net advantage of mixing PLA with TPSiU?
Response 3: The poor compatibility was demonstrated by DMA results. Nofar et al. proved that the increase of hard segment content in polyurethane can significantly improve the compatibility between PLA and TPU, but the use of TPU with lower hard segment was more conducive to enhancing the ductility and impact properties of the PLA. Due to the introduction of silicone segments in the polyurethane structure, the Si-O bond and polysiloxane as the main body which makes TPSiU have good flexibility. Therefore, compared with polyurethane, TPSiU has better toughening effect on PLA.
Point 4: Line 87: Could you give some further details regarding the time employed for the melt blend and injection-molding processes? Was extrusion performed only once?
Response 4: We have made corrections and supplements based on the reviewers’ comments.
Point 5: Figure 1: What are the y-axis units of this figure?
Response 5: The y-axis units of Figure 1 is W/g. And we have made correction according to the Reviewer’s comments.
Other changes:
In order to make the paper more complete, we have added some experimental data and revised the planning sequence of the third part of the paper.
We have studied reviewer’s comments carefully . We have tried our best to revise our manuscript according to the comments. Attached please find the revised version, which we would like to submit for your kind consideration.
We would like to express our great appreciation to the reviewers for comments on our paper. Looking forward to hearing from you.
Reviewer 4 Report
Presented work is interesting, but should be completed in some points:
- Introduction should be completed in the terms of thermoplastic silicone polyurethane elastomer and polymer blends where rubber phase is incorporated in rigid matrix
- Please, present the values of melt flow index for PLA and TPSiU
- Please indicate the shape and dimensions of specimens for tensile tests in section "2.3 Characterization"
- Please, present DSC thermogram, FTIR spectrum and TG/DTG curves for thermoplastic silicone polyurethane elastomer. In my opinion tensile properties of TPSiU should be also investigated
- DTG curves of prepared materials should be presented
- The values of elongation at break (from static tensile test) should be presented. Examplary stress-strain curves should be also presented.
- In Table 1 present the Tg values determined by DSC and DMTA method
- Obtained results should be discussed in the terms of chemical structure-morphology-properties relationship. Results can be discussed with other works were rubber phase is incorporated in rigid polymer matrix (like for example in HIPS)
- How to improve compatibility between PLA and TPSiU?
- Quality of Figures 1-7 should be improved (figures look like print screens, they should be exported as JPG or TIF images from the software)
Author Response
Response to Reviewer 4 Comments
Point 1: Introduction should be completed in the terms of thermoplastic silicone polyurethane elastomer and polymer blends where rubber phase is incorporated in rigid matrix
Response: As suggested by the reviewer, we highlighted the influence of soft segment (rubber phase) and hard segment (crystalline phase) in TPSiU on PLA in the introduction.
Point 2: Please, present the values of melt flow index for PLA and TPSiU
Response: Due to the reasons of raw materials and equipment, the values of melt flow index for PLA and TPSiU are temporarily unavailable. We are very sorry for this. Correspondingly, we have added rheological data for PLA/TPSiU. We appreciate for Editors/Reviewers’ warm work earnestly, and hope that the correction will meet with approval.
Point 3: Please indicate the shape and dimensions of specimens for tensile tests in section "2.3 Characterization"
Response: We have made correction according to the reviewer’s comments.
Point 4: Please, present DSC thermogram, FTIR spectrum and TG/DTG curves for thermoplastic silicone polyurethane elastomer. In my opinion tensile properties of TPSiU should be also investigated
Response: We have made corrections and supplements based on the reviewers’ comments. The crystalline properties, thermal properties and molecular structure of TPSiU were analyzed. And provides the tensile strength and elongation at break of TPSiU.
Point 5: DTG curves of prepared materials should be presented
Response: We have made corrections and supplements based on the reviewers’ comments.
Point 6: The values of elongation at break (from static tensile test) should be presented. Examplary stress-strain curves should be also presented.
Response: It is indeed as recommended by reviewer, so we provided the value of elongation at break and the stress-strain curve. We apologize for the negligence in this regard.
Point 7: In Table 1 present the Tg values determined by DSC and DMTA method
Response: We have made correction according to the reviewer’s comments. Tg values were determined by DSC.
Point 8: Obtained results should be discussed in the terms of chemical structure-morphology-properties relationship. Results can be discussed with other works were rubber phase is incorporated in rigid polymer matrix (like for example in HIPS)
Response: We re-edited the section 3 (Results and Disscussion) of the paper according to the reviewer’s suggestions, and discussed the results obtained from the relationship between chemical structure-form-property. And discuss the results with other work (polyurethane and natural rubber).
Point 9: How to improve compatibility between PLA and TPSiU?
Response: Some researchers have adopted the method of adding reactive compatibilizer or third component copolymer to enhance the compatibility. And Nofar et al. proved that the increase of hard segment content (crystalline phase) in polyurethane can significantly improve the compatibility between PLA and TPU, but the use of TPU with lower hard segment was more conducive to enhancing the ductility and impact properties of the PLA. For compatibility issues, we plan to use carbodiimide for reactive compatibilization in the next study.
Point 10: Quality of Figures 1-7 should be improved (figures look like print screens, they should be exported as JPG or TIF images from the software)
Response: We apologize for our negligence on the clarity of the picture. We have made correction according to the reviewer’s comments.
We have studied reviewer’s comments carefully . We have tried our best to revise our manuscript according to the comments. Attached please find the revised version, which we would like to submit for your kind consideration.
We would like to express our great appreciation to the reviewers for comments on our paper.
Round 2
Reviewer 1 Report
Reviewers' comments:
The authors revised the manuscript according to the reviewers' comments.
So that I recommended this manuscript accept for publication in Polymers.
Author Response
Dear Reviewers:
Thanks very much for your kind work and consideration on publication of our paper. On behalf of my co-authors, we would like to express our great appreciation to you.
Thank you and best regards.
Yours sincerely,
Keqing Han
hankeqing@dhu.edu.cn
Reviewer 2 Report
In line 147: The proper name for Czech is: Czech Republic.
In Lines 188-189: The melting peak, is an endothermic event.
Correct the number of Figures and their citations along the whole text.
Still remains the lack of comparison of their results with the literature reports, along the section 3. Need more explanation of the novelty of their work.
Author Response
Response to Reviewer 2 Comments
Point 1: In line 147: The proper name for Czech is: Czech Republic.
Response 1: We are very sorry for our incorrect writing. And we have made correction according to the Reviewer’s comments.
Point 2: In Lines 188-189: The melting peak, is an endothermic event.
Response 2: We are sorry that we made a mistake in the part about the melting endothermic peak. And we have made correction
Point 3: Correct the number of Figures and their citations along the whole text.
Response 3: We have checked (and corrected) the numbers of Figures and Tables.
Point 4: Still remains the lack of comparison of their results with the literature reports, along the section 3. Need more explanation of the novelty of their work.
Response 4: We are very sorry for our negligence of the reasons for choosing TPSiU. Polyurethane had been proven to have a better toughening effect. Due to the introduction of silicone segments in the polyurethane structure, the Si-O bond and polysiloxane as the main body which makes TPSiU have good flexibility and more excellent heat resistance. Therefore, compared with TPU, TPSiU has better toughening effect on PLA on the one hand, and better heat resistance on the other hand. And the experimental results obtained in the paper also proved this point. So we choosed TPSiU to plasticize PLA. The corresponding explanation has been added in the manuscript. And we have supplemented the comparison with other literature reports in 3.4 Compatibility and 3.5 Mechanical Properties.
We have studied reviewer’s comments carefully . We have tried our best to revise our manuscript according to the comments. Attached please find the revised version, which we would like to submit for your kind consideration.
We would like to express our great appreciation to the reviewers for comments on our paper.
Name: Keqing Han
E-mail: hankeqing@dhu.edu.cn
Reviewer 3 Report
Please include a statement in the final version of the manuscript for this reply
Response 2: Lactic acid can be divided into L-lactic acid and D-lactic acid, which are enantiomers of each other. The current mainstream method of synthesizing polylactic acid is to synthesize lactide from lactic acid first, and then form polylactic acid by ring-opening polymerization of lactide. In the preparation process of L-lactic acid, the production of D-lactic acid is inevitable. Therefore, the PLA reported in this study has only 98.6 wt% L-lactide content, and the other 1.4 wt% component is D-lactide content, which will not affect the experimental results.
Author Response
Response to Reviewer 3 Comments
Point 1:Please include a statement in the final version of the manuscript for this reply
Response 1: We have added this part to the manuscript according to the reviewer’s comments
Thanks very much for your kind work and consideration on publication of our paper. On behalf of my co-authors, we would like to express our great appreciation to you.
Thank you and best regards.
Yours sincerely,
Name: Keqing Han
E-mail: hankeqing@dhu.edu.cn
Reviewer 4 Report
- In my opinion the quality of Figures should be improved (The are indistinct, and should be exported as TIFFs or JPEGs from the software). For example: minor plots in Figure 7 (page 9) are not visible properly. Before you will resubmit a paper please save it as PDF and check the quality of Figures.
- Figure 2 should be removed if it the same spectrum as in the Figure 3.
- The curves from Figure 4 (page 5) should be introdcued into Figure 5 (page 6)
- The curves from Figure 4 (page 7) should be introduced into Figure 5 (page 7)
- Please, check (and correct) the numbering of Figures and Tables
- The values of elongation at break (in Table 3, page 9) should be supported by standard deviations
Author Response
Point 1: In my opinion the quality of Figures should be improved (The are indistinct, and should be exported as TIFFs or JPEGs from the software). For example: minor plots in Figure 7 (page 9) are not visible properly. Before you will resubmit a paper please save it as PDF and check the quality of Figures.
Response 1: We apologize for our negligence on the clarity of the picture. We have made correction according to the reviewer’s comments. And we have checked the clarity of the pictures one by one after converting the manuscript to PDF.
Point 2:Figure 2 should be removed if it the same spectrum as in the Figure 3.
Response 2: As Reviewer suggested, We have removed Figure 2.
Point 3:The curves from Figure 4 (page 5) should be introdcued into Figure 5 (page 6). The curves from Figure 4 (page 7) should be introduced into Figure 5 (page 7). Please, check (and correct) the numbering of Figures and Tables
Response 3: We are very sorry that we confuse the picture. We have checked and corrected the numbers of the Figures and Tables.
Point 4: The values of elongation at break (in Table 3, page 9) should be supported by standard deviations
Response 4: We have made corrections and supplements based on the reviewers’ comments.